



**Understanding soil loss in two permanent gully head cuts in the mollisol region of**
**Northeast China**
Chao Ma[1], Shoupeng Wang[1], Dongshuo Zheng[1], Yan Zhang[1], Jie Tang[2], Yanru Wen[3], Jie Dong[4]
1. School of Soil and Water Conservation, Beijing Forestry University, Beijing 100083, PR China.
2. Advanced Institute of Natural Sciences, Beijing Normal University at Zhuhai, Zhuhai 519087, China
3. Institute of Agricultural Resources and Regional Planning, Chinese Academy of Agricultural Sciences, Beijing
100081, China
4. Civil and Environmental Engineering Department, Clarkson University, NY, 13699, USA.
Corresponding Author: Professor Chao Ma, sanguoxumei@163.com
**Abstract:** Gravitational mass wasting on steep slopes plays an important role in permanent gully development. This
is typically driven by hydrological processes in the head cut and the hydromechanical response within the soil mass.
In this study, erosion intensities were observed in the head cuts of two permanent gullies in the mollisol region of
Northeast China during the rainy and snow melting seasons. To understand the physical process, soil water storage
and drainage capacity, and suction stress during the 111 d of the rainy season and 97 d of the snow melting season,
critical parameters such as soil moisture, temperature, and precipitation were investigated. This analysis also
examined the increase in pore water pressure, dissipation properties, and hydromechanical properties of the mollisols.
Under the same confining stress, the mollisols in the interrupted head cut of Gully No. II increased more rapidly and
dissipated pore water pressure more than at the uninterrupted head cut of Gully No I. The combination of the soil
water characteristic curve and the hydraulic conductivity function indicates that the mollisols of Gully No. II had a
higher air entry pressure and saturated hydraulic conductivity during the wetting and drying cycles than Gully No.
I. The head cut area of Gully No. II exhibited rapid water infiltration and drainage responses during rain events, with
a high soil water storage capacity during torrential rain, rainstorms, and snow melting seasons. The absolute suction
stresses within the mollisols of Gully No. II was lower than that in Gully No. I, which could lead to high erosion per
unit of steep slope area. Soil loss from gravitational mass wasting on steep slopes is closely related to soil suction
stress and we observed a correlation between erosion per unit gully bed area and the soil water storage. These
findings have deepened our understanding of the physical process of permanent gully development from the
perspective of the hydrological and hydromechanical behavior of gully head cuts.
**Keywords:** Gravitational mass wasting; Soil water characteristic curve; Erosion per unit area
**1 Introduction**
Gravitational mass wasting refers to the downward movement of rock, regolith, and/or soil caused by gravity
along the sloping top layers of the earth's surface (Evans, 2004; Allen et al., 2018). There are four types of mass
wasting, based on the speed of movement of the material and the level of moisture, namely, falls and avalanches,
landslides, flow, and creep (Bierman and Montgomery, 2014). They often occur in various sizes with undetermined
failure planes and are affected by hydrological and hydromechanical responses (Stein and LaTray, 2002; Rengers
and Tucker, 2014). On the steep slopes of permanent gullies, gravitational mass wasting involves debris-free soil
falling owing to bed undercutting driven by intensive channelized flow or persistent high soil moisture (Harmon and
Doe, 2001). Soil loss from gravitational mass wasting during the rainy season occurs when a steep slope loses support
from debris deposits. Meanwhile, soil loss during the melting season may result from persistent low soil suction
stress. In unsaturated soil mechanics, a high occurrence potential or intensive soil loss from gravitational mass
wasting corresponds to low soil suction stress (Lu and Godt, 2013). It remains unclear whether soil loss from
gravitational mass wasting corresponds to soil suction stress during these two stages.





Permanent gullies are initiated in locations where concentrated flows can erode and deliver bed sediments
(Kirkby and Bracken, 2009; Sidle et al., 2017) and expand when gravitational mass wasting occurs following instant
or constant water infiltration (Poesen et al., 2010; Tebebu et al., 2010). Permanent gully development can be
determined by the topographical threshold and volumetric retreat rate of gully head cuts (Svoray et al., 2012; Guan
et al., 2021; Zare et al., 2022), the gully length–area–volume relationship (Li et al., 2015 and 2017), and their function
in the upstream drainage area and rainy days in different environments (Hayas et al., 2019). Soil loss from permanent
gullies is largely influenced by hydrological factors (Gómez-Gutiérrez et al., 2012), such as the flow rate, total water
volume, rainfall intensity and amount, and hydromechanical properties of the soil mass. Soil properties are affected
by land use, plant roots, texture, and structure. The hydrological process near the head cut, the hydromechanical
response of soil mass in reaction to water infiltration, and their relationship with soil loss from gravitational mass
wasting remain unknown. Under natural conditions, water infiltrates either following rain events or snow/ice melting
events. The infiltration rate strongly depends on both the amount and intensity of precipitation, which leads to soil
water storage. However, the amount of stored water varies owing to the amount of rainfall and the melting rate or
temperature. During the snow/ice-melting season, the duration of water infiltration persists longer than that of rain
events because of prolonged soil saturation and an extended period of low soil suction stress. This may generate
more soil loss owing to gravitational mass wasting. However, rain events typically generate intensive channelized
flows, which erode steep slopes and trigger gravitational mass wasting. Therefore, it is challenging to compare soil
loss in the two seasons. However, this issue could be addressed by considering the associated hydrological processes
of head cuts and hydromechanical responses within the soil mass.
In the mollisol region of Northeast China (MEC), over 296,000 permanent gullies have developed since 1960
(Yang et al., 2017; Dong et al., 2019). Gravitational mass-wasting processes have caused rapid gully widening due
to overfarming and a lack of maintenance (Wang et al., 2009). Various studies have focused on the hydrological
processes affecting ephemeral gully development and volume disparities caused by rain/snow melting (Tang et al.,
2022; Jiao et al., 2023), tillage practices (Xu et al., 2018; Li et al., 2021), and morphology (Zhang et al., 2016).
Permanent gullies pose a greater threat to croplands than ephemeral gullies because the soil loss from permanent
gully erosion can be as high as 50–65% of the total loss (Zhang et al., 2022). The relatively high area increasing
ratio is affected by the combination of permanent gullies with cropland use, a large ridge orientation angle, and a
sunny orientation (Li et al., 2016; Liu et al., 2023). Tang et al. (2023) provided evidence of the rainfall threshold for
permanent gully development and found that the maximum value of 3 d accumulative rainfall best explained
permanent gully bed erosion, and the cumulative value of erosive rainfall best accounted for gravitational mass
wasting. However, gravitational mass wasting on the steep slope of a permanent gully can occur either during the
rainy season or snow melting season (Zhang et al., 2020; Zhou et al., 2023). Currently, the hydrological processes
near the head cut and the hydromechanical response of mollisols to water infiltration in the two seasons have never
been documented, and the associated soil loss from gravitational mass wasting is poorly understood. In the MEC,
although the duration of the snow/ice melting season is shorter than that of the rainy season (Wang et al., 2021; Fan
et al., 2023; Went et al., 2024), the time for snow melting water or rainy water infiltration and the soil water storage
and drainage processes are significantly different. Soil water storage may exceed drainage because of continuous
meltwater infiltration and limited water drainage paths. Rain infiltration during the summer season temporarily
increases and then decreases once the rain event ceases and the water drains. Stored water significantly depends on
rainfall events (Farkas et al., 2005; Xu et al., 2018). Therefore, the duration of low soil suction stress, such as high
soil moisture, differed substantially between the two seasons. Another effect is channelized water during intensive
rainstorms (Wen et al., 2021), which may erode the bed and result in gravitational mass wasting. Therefore, the soil
loss from gravitational mass wasting may coincide with the soil suction stress in the snow/ice melting season.
Meanwhile, this coincidence may not exist in the rainy season.





Soil loss from gravitational mass wasting on the steep slope of a permanent gully is poorly understood in the
MEC. To date, relatively few studies have addressed its relationship with the suction stress of the soil mass. This
work has focused on soil loss during the rainy and melting seasons in the head cuts of two permanent gullies, where
one head cut experiences no human activity, whereas the other does. Soil loss in the head cut area during the rainy
and melting seasons was observed. The differences in the physical properties of the mollisols, such as pore water
pressure dissipation at a given confining stress, the soil water characteristic curve (SWCC), and the hydraulic
conductivity function (HCF), were compared. The soil loss per unit area on the steep slope and gully bed was
analyzed for the soil water storage and drainage and the hydromechanical response, respectively. The results of this
study deepen our understanding of permanent gully expansion using classical mechanics.
**2 Study area**
Northeast China is one of the three main mollisol regions worldwide, with a total area of 1,030,000 km². It
contributes 20% of the grain and more than 40% of the corn in China. Most of the mollisol region was gradually
converted from native vegetation to cropland beginning in the late 19th Century. Croplands constitute 80% of the
total land area, and the main crop types are soybean and corn. The study area is located in the typical heavy gully
erosion area of the mollisol region of Northeast China, where native grasslands and forests have been fully converted
into croplands since 1968. It is situated in a transitional rolling hilly area extending from the Songnen Plain to the
Greater Khingan Mountains in the west, the Lesser Khingan Mountains in the north and near the Nen River (Fig.
1a). Owing to the gentle landscape, the farmland in the study area is covered by a thick black organic soil layer, with
sandstone, mudstone, and sandy conglomerate underneath.
The two permanent gullies examined in this work are 1.4 km apart and are located on the south-facing and
north-facing rolling slopes, respectively (Figs. 1b and 1c). The catchment area above Gully No. I is 0.22 km². The
relative relief and channel gradient are 25.85 m and 3.3%, respectively. The catchment above the head cut of Gully
No. II is 0.35 km², and the relative relief and channel gradient are 26.1 m and 3.2%, respectively. The width of Gully
No. I gradually broadened, whereas Gully No. II narrowed and Gully No. I was deeper (Figs. 2a and 2b). The mean
depth of the Gully No. I was 3.5 m while that of Gully No. II was 1.23 m. The mean length and width of No. I gully
were 25.3 m and 8.72 m, whereas those of Gully No. II were 28.2 and 5.61 m. The gully area for No. I was 199.3
m² and the volume was 863.6 m³. For Gully No. I, the area and volume of the gully were 143.3 m² and 123.6 m³.
The two gullies are still expanding because they are connected to the river network, which drains water into the
Nen River. Although grass covers the area near the sidewall and ridge along the gully, mass wasting movement
frequently occurs during the melting and rainy seasons. The differences in the gully planform and depth indicate that
the mass movement at the sidewall or head cut has distinctive rates and scales. The mass movement at the sidewalls
of the two gullies differed in scale, as shown in Figs. 2c and 2d. The height and width of the Gully No. II were lower
than those of the Gully No. I (Fig. 3). The head cut area of Gully No. II underwent tillage activities, whereas the
head cut area of Gully No. I has not. Therefore, Gully No. II is representative of the initial development stage for a
large permanent gully.
The study area has a continental monsoon climate with variable annual precipitation ranging from 347 to 775
mm, with an average of 546 mm between 1971 and 2018 (Tang et al. 2023). Rainfall mainly occurs between June
and August, accounting for 70–90% of the annual precipitation, with an average of 461 mm. Snowfall occurs mainly
from November to April, accounting for 10–30% of the annual precipitation. The average temperature in the coldest
and warmest months are –22.5 °C and 20.8 °C, respectively, with an annual average temperature of 0 °C.



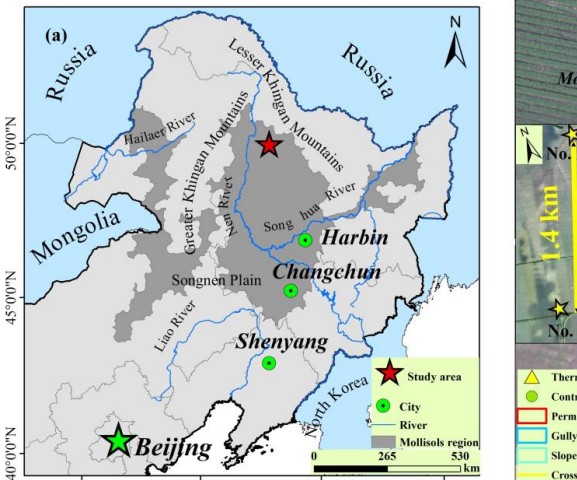

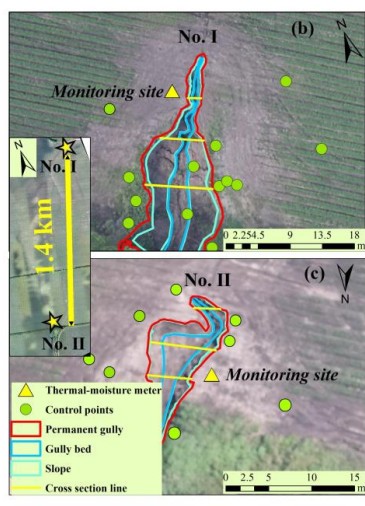

**Fig. 1.** Location of the two permanent gullies in the mollisol region of Northeast China. **(a)** The red star marks the observation site in the study area (from ESRI). **(b)** Monitoring sites and ground controlling points at permanent Gully No. I. **(c)** Monitoring sites and ground controlling points at permanent Gully No. II. (background of **a** is from ESRI. The area between the blue lines marks the gully bed, and the area between pink and blue lines marks the steep slope.

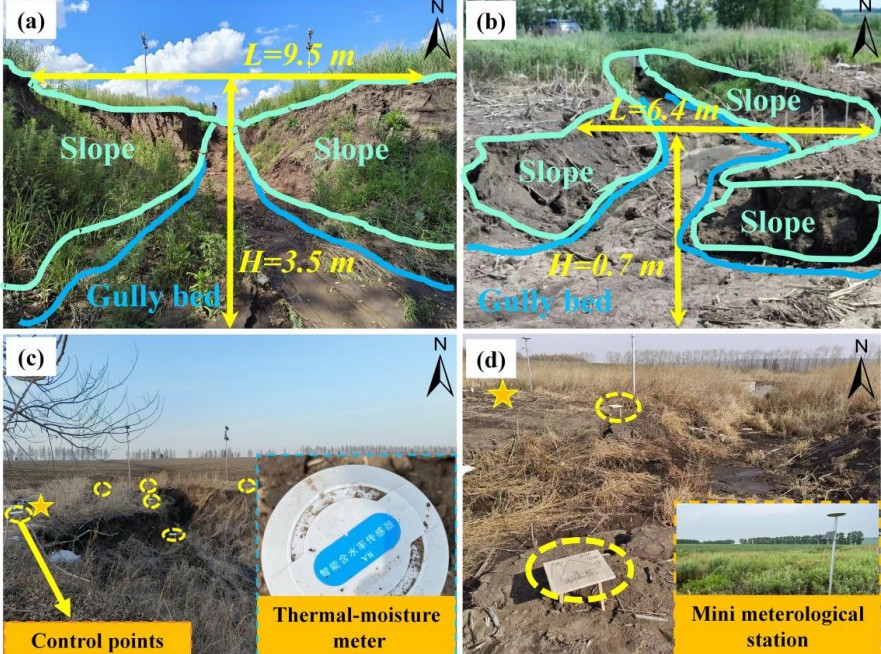

**Fig. 2.** A close view of the steep slope and head cut of the two permanent gullies, with **(a)** cross-section and upstream view of the permanent Gully No. I, **(b)** cross-section and downstream view of the permanent Gully No. II, **(c)** ground control points (blue dot circles) and the soil moisture–temperature monitoring site (yellow star) at





permanent Gully No. I, and **(d)** ground controlling points and the soil moisture–temperature monitoring sites at
permanent Gully No. II. The location of the head cut of the two gullies is shown in Fig. 1. The area between
the blue lines marks the gully bed. The area between the pink and blue lines marks the slope.

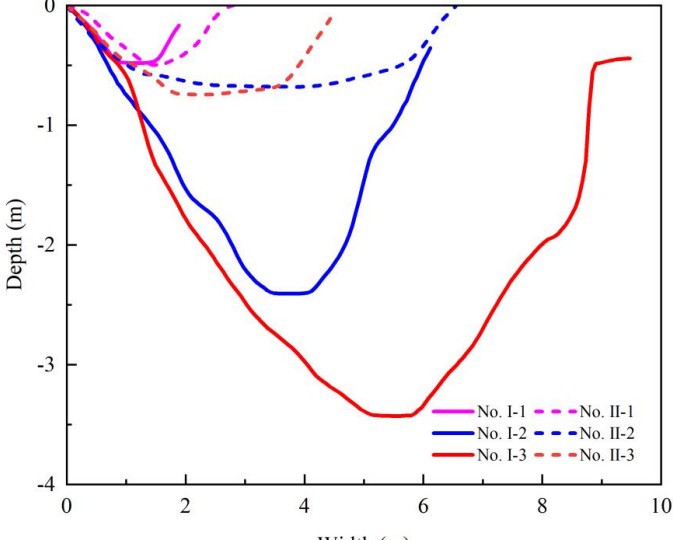


**Fig. 3.** Difference of the two permanent gullies' cross-section. The location of the cross-section lines is shown in
Figs. 1b and 1c.

**3 Material and methods**
**3.1 Monitoring work**

Near the gully head cut, frequency–domain reflectometry sensors were installed to monitor the soil moisture
and air temperature at depths of 20, 40, 60, and 80 cm (Fig. 2c). These two monitoring sites share the same rainfall
records as Gully No. II (Fig. 2d). A trench was dug to obtain soil samples from the two monitoring sites. The soil
samples were used for pore water pressure dissipation tests via consolidated undrained triaxial compression tests
(CU) using a GDS triaxial apparatus (GDS, UK), and the unsaturated permeability was measured using the transient
release and imbibition method (TRIM; Lu and Godt, 2013).

To observe the gravitational mass wasting process during the rainy and melting seasons, the study area was
scanned using numerous control points (the dots in Figs. 1a and 1b and dashed circles in Figs. 2c and 2d) installed
in and around the gully area, and an unmanned aerial vehicle (UAV) was used. These control points were used to
improve the accuracy of the UAV-derived map and digital elevation model to obtain highly accurate topography
data. Three flights on June 28, 2022, October 17, 2022, and June 20, 2023, were performed with the same flight
routine and image overlap. The two frontier flights in 2022 spanned 111 d during the rainy season. The latter two
covered the winter of 2022 and spring of 2023. As low soil moisture persists from October each year and snow cover
in the winter season does not result in gravitational mass movement, the effective duration of the melting season
starts on March 15, 2023. Therefore, the melting season in this study lasted 97 d. We used Pix4D software to process
the image synthesis and gully topography production, which can reallocate the point cloud and filter the points of
the vegetation layer. As the points of the vegetation layer, mainly the grass blades, are changeable in plant height,
whereas the ground point is fixable, the vegetation layer can be filtered out and removed using the filtering tool. The





DEM products were spatially registered in ArcGIS 10.2 using a standard layer of orthoimages, ground control points,
and spline functions (Table 1). The erosion depth of the head cut was then obtained from the difference between the
two DEMs. Therefore, the linearity and erosion per unit area could be calculated using the erosion depth and grid
size. The differences between the two digital elevation models generated positive and negative terrain, which showed
soil loss from gravitational mass wasting. The eroded soil volume in a unit of steep slope surface area, termed erosion
per unit area, was applied to address the erosion caused by gravitational mass wasting.
**Table 1.** Detailed information of three UAV flights and the digital elevation models

| UAV model | Flight date | Season/ duration | Flight height (m) | DEM accuracy (m) | Image overlap (%) |
|---|---|---|---|---|---|
| DJI Inspire 2 RTK | 2022.06.28 | / | 200 | 0.058 | 80 |
| DJI Phantom 4 RTK | 2022.10.17 | Rainy/111 d | 500 | 0.108 | 80 |
| DJI Phantom 4 RTK | 2023.06.21 | Melting/97 d | 150 | 0.042 | 80 |

**3.2 Tests of pore water pressure rising and dissipation**
The consolidation module of the GDS triaxial apparatus was used to record the pore water pressure within the
soil mass under a given confining stress. The soil samples were initially saturated in a vacuum pump and then
consolidated in the chamber of the GDS apparatus at effective confining pressures of 100, 200, and 300 kPa with a
10-kPa backpressure. The consolidation process was completed when the pore water pressure decreased to the
backpressure values.
For the pore water increasing stage:
$$P_\uparrow = P_0 \times t^{b_\uparrow}\qquad(1)$$
where $P_\uparrow$ is the recorded pore water pressure during the increasing stage (kPa), $P_0$ is the initial pore water pressure
since loading (kPa), $t$ is the time (s), $b_\uparrow$ is the rising proxy reflecting the steepness of the power-law curves of pore
water pressure increase.
For the pore water dissipation stage:
$$P_\downarrow = \frac{P_{max}}{1+b_\downarrow \times t}\qquad(2)$$
where $P_\downarrow$ is the recorded pore water pressure during the dissipation stage (kPa), $P_{max}$ is the maximal pore water
pressure since loading (kPa) and is the rollover point in the pore water pressure curve, $t$ is the time (s), $b_\downarrow$ is the
dissipation proxy reflecting the water drainage ability of soil mass at given confining pressure and reflects the
concavity of the pore water pressure dissipation curve.

**3.3 Hydromechanical properties**
TRIM was used to test the unsaturated permeability of the soil mass (Lu and Godt, 2013). The SWCC and HCF
were obtained using Hydrus 1-D (Wayllace and Lu, 2012). Using the models proposed by Mualem (1976) and van
Genuchten (1980), the constitutive relations between the suction head ($h$), water content ($\theta$), and hydraulic
conductivity ($K$) under drying and wetting states can be represented by the following equation:
$$\frac{\theta-\theta_r}{\theta_s-\theta_r} = \left[\frac{1}{1+(\alpha|h|)^n}\right]^{1-\frac{1}{n}}\qquad(3)$$
and
$$K = K_s \frac{\left\{1-(\alpha|h|)^{n-1}[1+(\alpha|h|)^n]^{\frac{1}{n}-1}\right\}^2}{[1+(\alpha|h|)^n]^{\frac{1}{2}-\frac{1}{2n}}}\qquad(4)$$





where $\theta_r$ is the residual moisture content (%), $\theta_s$ is the saturated moisture content (%), $\alpha$ and $n$ are empirical
fitting parameters, $\alpha$ is the inverse of the air-entry pressure head, $n$ is the pore size distribution parameter, and $K_s$
is the saturated hydraulic conductivity (cm/s).
Based on the observed volumetric water content and the SWCC, the suction stress ($\sigma^s$, kPa) throughout the
observation stage can be expressed as:
$$\sigma^s = -\frac{S_e}{\alpha}\left(S_e^{n/(1-n)} - 1\right)^{1/n} \tag{5}$$

**3.4 Soil water storage and drainage**
In this study, the hydrological process of the steep slope is of utmost importance for analyzing gravitational
mass wasting because of the varied soil water storage and drainage in the rainy and snow-melting seasons. Soil water
is temporally stored during rainstorms, but drains after the rainstorms cease. The drainage process during melting is
not addressed herein because melting water constantly contributes to high soil moisture. Therefore, soil water storage
($S_s$) during rainstorms and the snow-melting season and drainage ($S_d$) after a rainstorm ceases can be evaluated
using the soil depth and the difference between the maximum soil moisture and antecedent soil moisture:
$$S_e = \frac{\theta - \theta_r}{\theta_s - \theta_r} \tag{6}$$

$$S_s = S_e^w \Delta h_i \tag{7}$$

$$S_d = P - S_e^d \Delta h \tag{8}$$

where $S_e$ is the degree of saturation, $\theta$ is the in-situ observed volumetric moisture content measured (%), $\Delta h_i$ is
the soil layer $i$ (200 mm in this work, $i$ = 1, 2, 3, 4), $S_e^w$ and $S_e^d$ are the residual soil moisture in the wetting and
drying processes (%), and $P$ is the accumulated rainfall (mm) and equals to 0 mm in the snow-melting season. To
show the soil water storage during the rainy and snowmelt seasons, and the water drainage after rainfall, all the
information was considered, including rainfall amount, air temperature, soil moisture, and temperature in various
soil layers. The recorded rain events were categorized into four groups, that is, light rain, moderate rain, torrential
rain, and rainstorms, with rain amounts of < 10, 10–25, 25–25, and 50–100 mm, respectively.
**4. Results**
**4.1 Erosion per unit area of gully bed and slope**
The erosion per unit area in both bed and slope areas during the snowmelt season for Gully No. I was greater
than that in Gully No. II (Fig. 4). This could have been driven by the low meltwater storage and high meltwater
runoff at the head cut of Gully No. I. During the rainy season, the erosion per unit area for bed of Gully No. II was
greater than that of Gully No. I. This may have resulted from rapid soil water storage and drainage producing
intensive runoff at the head cut of Gully No. II. The erosion of steep slopes is mainly due to gravitational mass
wasting. For Gully No. II, erosion per unit area during the snowmelt season was significantly greater than that during
the rainy season. During the snow melting season, the erosion per unit area for the slope in Gully No. II was greater
than that in Gully No. I. Although erosion per unit area during the rainy season for Gully No. I was higher than that
of Gully No. II, the difference was negligible compared to that in the snow-melting season. The slopes of the
permanent gully were steep, and the stability of the slope primarily depended on the soil suction stress as a function
of the hydromechanical properties and the soil moisture.
As the channel bed erosion was closely correlated with the hydrological process and the slope erosion
corresponded to the soil suction stress, further examination of the associated soil water storage and drainage and the
hydromechanical properties of the soil mass in the two permanent gullies was conducted. One of the differences in
the hydrological processes in the head cut indicates that soil water storage and drainage occur during the rainy season.
Water drainage was absent during the snowmelt season. These results are due to the continuous melting of water





from snow and ice in macropores and fissures. Once the melting process was completed, the soil water storage
process ceased with the onset of the water drainage process during the transition time between the snow melting and
rainy seasons.

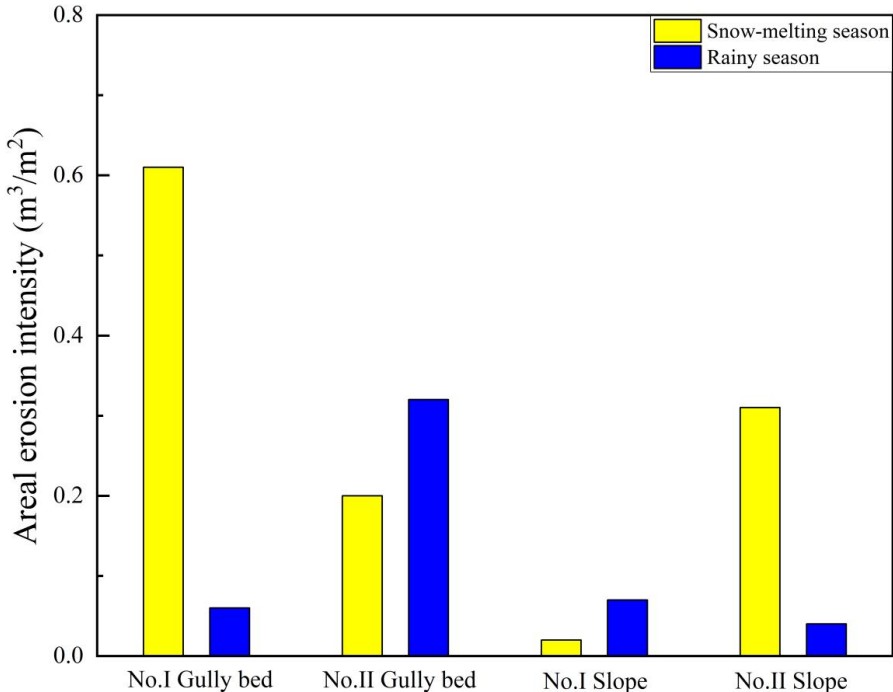

**Fig. 4.** Differences in the erosion per unit area for the gully bed and slope
**Table 2.** The physical properties and pore water pressure changes in the soil mass

| Parameters | Definition | Confining pressure (kPa) | Permanent gully No. I | Permanent gully No. II |
|---|---|---|---|---|
| $v\uparrow$ (kPa/min) | Pore water rising ratio | 100 | 11.83 | 23.04 |
| | | 200 | 4.86 | 90.52 |
| | | 300 | 5.55 | 10.92 |
| $b\uparrow$ | Pore water rising proxy as Eq. (1) | 100 | 0.23 | 0.25 |
| | | 200 | 0.24 | 0.46 |
| | | 300 | 0.30 | 0.41 |
| $v\downarrow$ (kPa/h) | Pore water dissipation ratio | 100 | 3.68 | 22.77 |
| | | 200 | 3.32 | 194.47 |
| | | 300 | 3.66 | 23.94 |
| $b\downarrow$ ($\times10^{-5}$) | Pore water dissipation proxy as Eq. (2) | 100 | 9.97 | 79.70 |
| | | 200 | 7.80 | 79.40 |
| | | 300 | 6.82 | 18.10 |
| $c$ (kPa) | Effective cohesion | | 11.3 | 7.2 |
| $\varphi$ (°) | Effective friction angle | | 16.3 | 21.3 |
| $\gamma$ (kN m$^{-3}$) | Unit weight | | 14.1 | 12.5 |




### 4.2 Physical properties of mollisols

**4.2.1 Pore water pressure rising and dissipation**

Under the same confining pressure, pronounced differences were observed in the rising and dissipation ratios of the pore water pressure within the mollisols of the two gullies. The pore water pressure results during the consolidation process at effective confining pressures of 100, 200, and 300 kPa were compared (Fig. 5). The physical properties, and the rising and dissipation ratios and proxies are listed in Table 2. The peak value of the pore water pressure within the mollisols of Gully No. I was higher than that in Gully No. II. The peak value of the pore water pressure within the mollisols of Gully No. II increased to 57.6, 139.0, and 141.7 kPa under the confining stresses of 100, 200, and 300 kPa, respectively. In contrast, the peak value of the pore water pressure within the mollisols of Gully No. I increased to 87.9, 176.1, and 237.3 kPa, respectively.

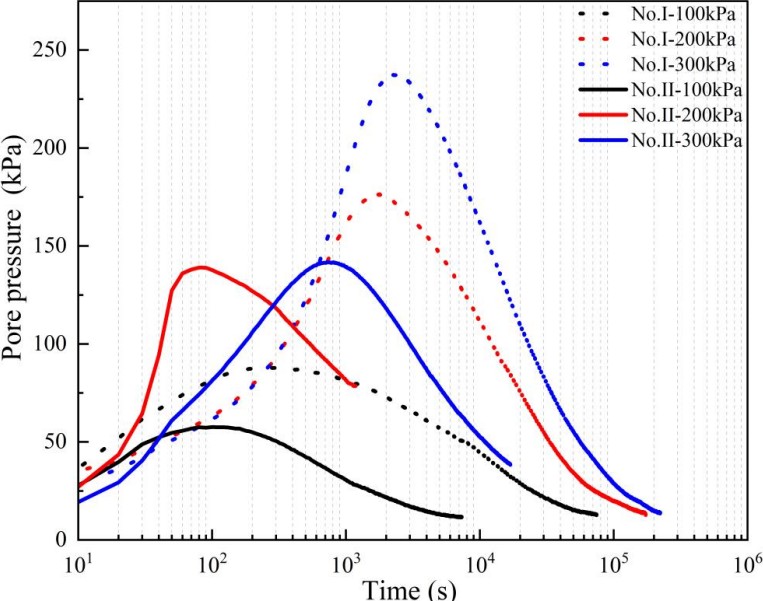

**Fig. 5.** Variation in pore water pressure under effective confining pressure of 100, 200, and 200 kPa by GDS triaxial shear tests (GDS Instruments, UK). The proxy for the pore water pressure rising and dissipation are calculated using Eqs. (1) and (2). The rising and dissipation ratio is calculated using the pore water pressure difference during a given time interval. The values of proxy and ratio are shown in Table 2.

The high peak pore water pressure illustrates that the mollisols of Gully No. II had strong hydraulic conductivity as the ratio increased, and the proxy and dissipation ratio and proxy represented the pore connectivity. During the rising stage, the rising ratio of the mollisols in Gully No. II was 2 to 18.6 times greater, and its rising proxy was 1.08 to 1.92 times larger than that of Gully No. I. Within the dissipation stage, the ratios were 6.20 to 58.6 greater, and its proxies were 2.65 to 8.0 times larger than those for mollisols of Gully No. I. The largest difference between these two gullies was observed under a confining stress of 200 kPa. Therefore, the increase in the pore water pressure and dissipation properties suggests that the head cut of Gully No. II may have exhibited active hydrological processes.

**4.2.2 Hydromechanical properties of mollisols**



Fig. 6 shows the results of the TRIM tests, SWCC, HCF, and estimated suction stress with varying degrees of
saturation. The water outflow mass was measured at 10-min intervals during the drying and wetting processes. The
water outflow masses measured for the mollisols in Gully No. II were generally higher than those of the mollisols
in Gully No. I. For the drying tests using mollisols from Gully No. II and No. I, the water outflow masses were
0.0713 and 0.060 g per 10 min, respectively. For the wetting tests, the water outflow masses were 0.031 and 0.0208
g per 10 min, respectively (Fig. 6a). Overall, the permeability of mollisol Gully No. II was higher than that of mollisol
Gully No. I. The same results were obtained for the pore water pressure increase, dissipation ratio, and proxy, as
shown in Table 2.

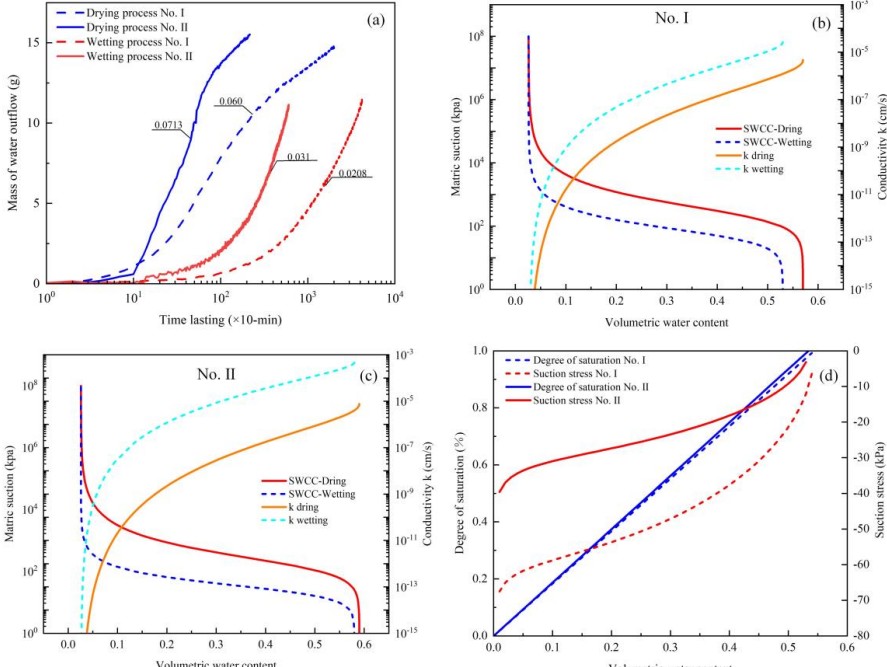


**Fig. 6.** Differences in the hydromechanical properties of the two soil masses. **(a)** Water flow mass in the drying and
wetting process. **(b)** SWCC for soil mass of permanent Gully No. I. **(c)** SWCC for soil mass of permanent Gully
No. II. **(d)** Suction stress–volumetric water content curves for the two soil masses. The mass of water outflow
was recorded at 10 min for each test.


Using the parameters listed in Table 3, the SWCC and HCF curves of the mollisols were plotted (Figs. 6b and
6c). Air entry pressure and residual water content are two important parameters that describe the hydrological and
mechanical characteristics of mollisols. The air entry pressure represents the critical value at which air enters the
saturated soil and begins to drain. In comparison, the values of $\alpha^d$ and $\alpha^w$ together prove that the required air entry
pressure for mollisols in Gully No. I was greater than that in Gully No. II, and the differences were 79.4 kPa and
28.0 kPa under drying and wetting conditions, respectively (Table 3). Therefore, water infiltration into Gully No. II
during either the rainy or snow melting seasons was more active than in Gully No. I. The residual moisture did not
vary markedly owing to the similarity in the soil types.
The saturated hydraulic conductivities of the mollisols in Gully No. I were lower than those in Gully No. II in
both the drying and wetting processes. In Table 2 and Fig. 5, the pore water pressure rising ratio and proxy and the





dissipation ratio and proxy further indicate that the permeability of the mollisols in Gully No. II was higher than that
in the mollisols of Gully No. I. Therefore, the pore water pressure changed with varying confining stress, air entry
pressure, and saturated hydraulic conductivities under drying and wetting conditions, suggesting that it is more
challenging for the mollisols in Gully No. I to absorb and drain more water compared to mollisols in Gully No. II.
Figs. 6b and 6c present the matric suction and hydraulic conductivity at various soil moisture levels. However,
it was not possible to compare the level of suction stress with various hydrological and mechanical parameters, as
listed in Table 3. Hence, the suction stress at various soil moisture levels was determined (Fig. 6d). The absolute
suction stress at the specified soil moisture for mollisols in Gully No. I was higher than that of mollisols in Gully
No. II, suggesting a higher possibility of gravitational mass wasting for the mollisols in Gully No. II.

**Table 3.** Parameters describing the SWCC and the HCF from Hydrus 1D.

| Parameters | Definition | Permanent gully | |
|---|---|---|---|
| | | No. I | No. II |
| $\theta_r$ | Residual moisture | 0.0262 | 0.0259 |
| $\theta_s^d$ | Saturated moisture | 0.57 | 0.59 |
| $\theta_s^w$ | | 0.53 | 0.58 |
| $\alpha^d$ (kPa$^{-1}$) | The inverse of the air entry pressure head | 0.0042 | 0.0063 |
| $\alpha^w$ (kPa$^{-1}$) | | 0.0183 | 0.0375 |
| $n^d$ | The pore size distribution parameter | 1.69 | 1.68 |
| $n^w$ | | 1.95 | 1.91 |
| $K_s^d$ (cm s$^{-1}$) | Saturated hydraulic conductivity | $4.73 \times 10^{-6}$ | $7.82 \times 10^{-6}$ |
| $K_s^w$ (cm s$^{-1}$) | | $2.64 \times 10^{-5}$ | $4.26 \times 10^{-4}$ |

Notes: the superscript $d$ and $w$ indicate drying and wetting states.

**4.3 Hydrological response**
**4.3.1 Monitoring results**
In total, 24 light rain events, two moderate rain events, five torrential rain events, and one rainstorm event were
recorded (Fig. 7a). During the snow melting season, the air temperature started to increase above 0 °C on March 20
with an increasing gradient of 0.15 °C per day, which reached 2.3 °C per day after April 23 (Fig. 7b). For soil
moisture changes, the volumetric water content at a depth of 20 cm for Gully No. II greatly increased from April 23,
whereas it only slightly increased for Gully No. I. This suggests that the head cut of the Gully No. II may have
experienced higher soil moisture levels. Soil moisture throughout the rainy and snowmelt seasons had dissimilarities
between sites. During the rainy season, the volumetric water content at a depth of 20 cm persistently remained at a
lower level of soil moisture than at the other three soil depths, as shown in Fig. 7c. However, during the snow melting
season, the volumetric water content of the 40 cm soil layer was the highest (Fig. 7d). Overall, the soil moisture
content of Gully No. II, in both the rainy and snowmelt seasons, exhibited greater fluctuations than Gully No. I.
Water infiltration from rain events or snowmelt into the head cut of Gully No. II was more active than that of Gully
No. I. The observed difference proves that the stored and drained water from the head cut of Gully No. II was
significantly greater than that in Gully No. I.



To further analyze the differences in water infiltration during the rainy and snowmelt seasons, the rate of soil moisture increase at a depth of 20 cm was compared in detail (Fig. 8). Among the four types of rain events, the mean rate of increase for Gully No. II were 0.027, 0.053, 0.102, and 0.356, respectively, which were 1.12, 1.35, 1.34, and 1.78 larger than those for Gully No. I (Figs. 8a and 9a). During the snow melting season, the soil moisture increase ratios in the initial, medium, and final stages for Gully No. II were 3.48, 1.60, and 1.66 times, respectively, than those in Gully No. I (Fig. 8b). Therefore, the water infiltration ratios for the head cut areas of Gully No. II during the rainy and snowmelt seasons.

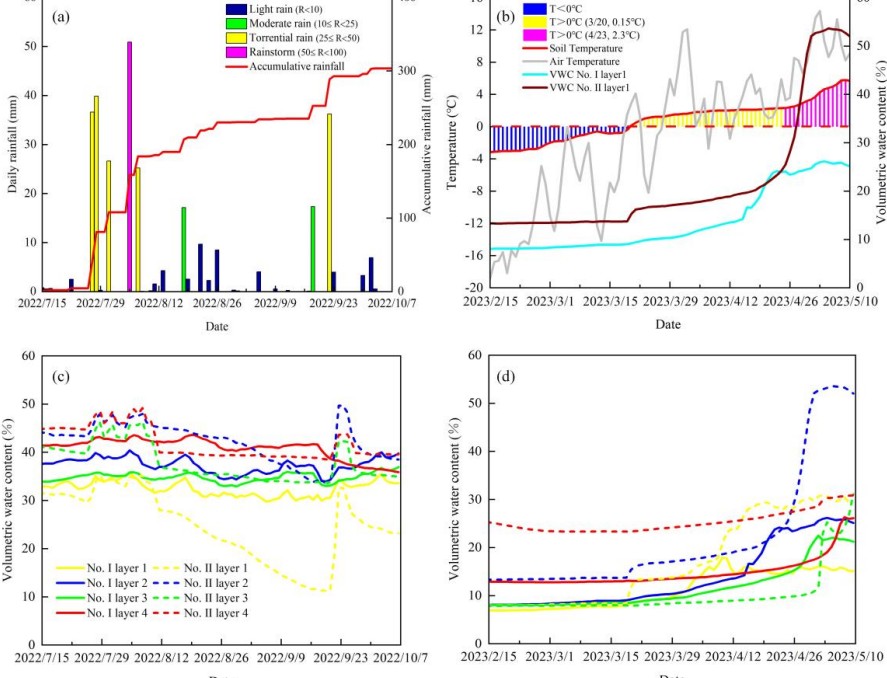

**Fig. 7.** Field-monitored rainfall conditions, air and ground temperature, and volumetric water content. **(a)** Rain events during the rainy season. **(b)** Soil, air temperature, and volumetric water content during the snow melting season. **(c)** and **(d)** Monitored volumetric water content during the rainy season and snow melting season.

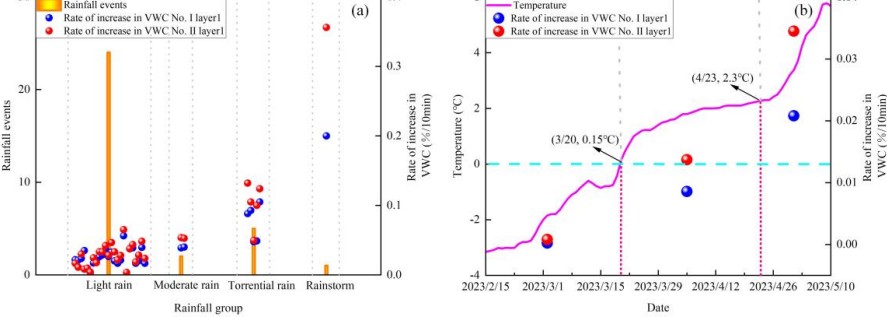

**Fig. 8.** Volumetric water content increasing ratio in snow melting ratio and the rainy season. **(a)** Rate of increasing of VWC at varied rain events. **(b)** Rate of increase in VWC at three stages of temperature increase.

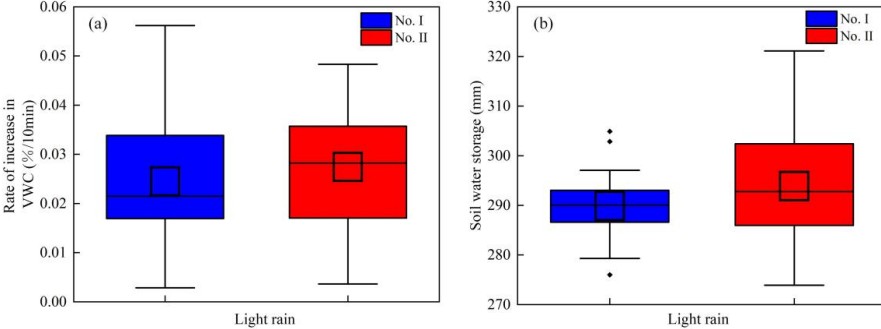

**Fig. 9.** Hydrologic behavior for gully head cut during light rain events. (**a**) Lower rate of increase in VWC for Gully No. I. (**b**) Higher soil water storage for Gully No. II. The three crossing lines of the boxes show the 75th quantile ($Q_3$), median ($Q_2$), and 25th quantile ($Q_1$) from top to bottom. The length of the box is referred to as the interquartile range (IQR = $Q_3 - Q_1$). The crossed square inside the box is the average value. The upper limit and lower limit of whiskers are $Q_3$+1.5IQR and $Q_3$−1.5IQR, respectively. The solid squares are the outliers.

**4.3.2 Soil water storage and drainage**

Fig. 10 shows the stored and drained water in the soil column at the head cuts of the two gullies. During the snowmelt season, the water stored in Gully No. II was higher than that in Gully No. I. The stored water ratio was calculated by dividing the amount of water stored in Gully No. II based on the amount stored in Gully No. I was typically larger than 1.0 throughout the snowmelt season (Fig. 10a). This ratio increased abruptly from April 26. Therefore, the amount of water stored in the head cuts of Gully No. II was higher.

Regarding the four types of rain events, the mean stored water for the head cuts of Gully No. II during the 24 light rain events was greater than that in Gully No. I (Figs. 9b and 10b). The differences in water stored in the head cuts of the two gullies were 4.0, 8.1, 15.2, and 46.3 mm, respectively. Therefore, the stored water, either in the snow melting season or rainy season, was higher in the head cuts of Gully No. II. However, the water stored in the head cuts of Gully No. II was not always higher than that in Gully No. I, as shown in Fig. 10c. From August 26 to September 3, 2022, the water stored at the head cut of Gully No. II was lower than that in Gully No. I. This could be attributed to high temperatures and light rain events. However, the water stored in the head cuts of Gully No. II exceeded that of Gully No. I during a torrential rainfall event on September 22. The soil water storage capacity of Gully No. II has stronger fluctuations. Rapid water infiltration generally occurs with rapid water drainage. Fig. 10d shows the water drainage and drainage ratios of the two gullies during the rainy season. The water drained from Gully No. II was higher than that in Gully No. I. Therefore, the head cut area of the Gully No. II had better soil water storage capability in snowmelt and rainy seasons and more rapid water drainage in the rainy season than Gully No. I.

In summary, rapid soil water storage and drainage for the head cuts of Gully No. II during torrential rain or rainstorms coincided with both the observed pore water pressure rise and dissipation and the hydromechanical properties of mollisols. The high permeability of mollisols at the head cut of Gully No. II was attributed to more rapid soil water storage, drainage processes, and stored water. This could have a considerable influence on the erosion intensity of the steep slope and gully bed of the permanent gully.



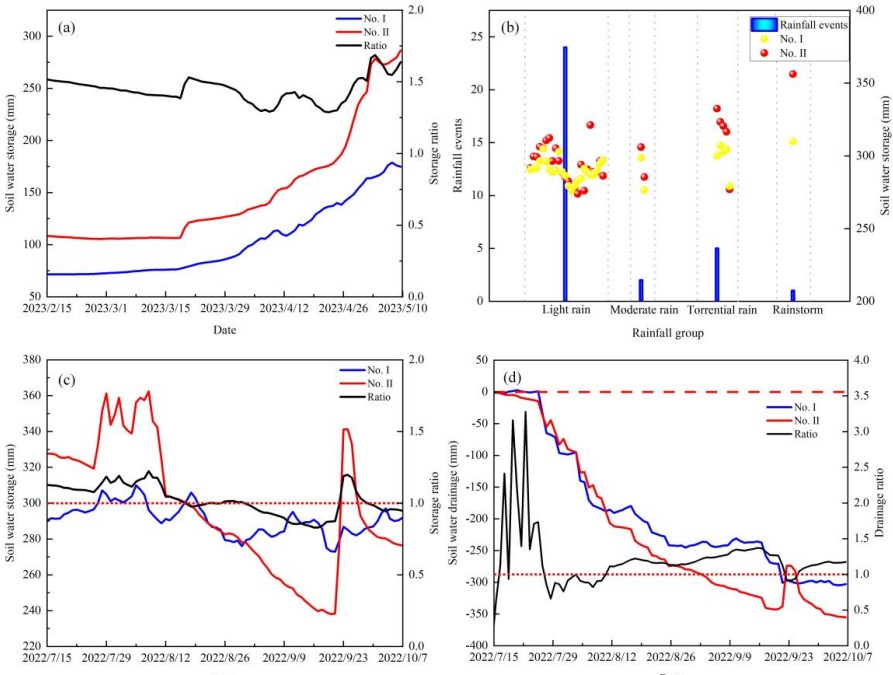

**Fig. 10.** Hydrological response during the rainy and snow melting season. **(a)** Soil water storage and the storage ratio during the snow melting season. **(b)** Soil water storage at varied rain events. **(c)** Soil water storage and the storage ratio for the two permanent gullies. **(d)** Soil water drainage and the drainage ratio during the rainy season. During the rainy season, soil water storage and drainage synchronously change with the onset and end of rainfall.

**4.4 Hydromechanical response and soil loss**

As the mollisols in the head cut area of the two permanent gullies differed in hydromechanical properties, the monitored soil moisture varied greatly in the field. The suction stress was estimated according to the field-monitored soil moisture at each site and the relationship between the soil moisture and matric suction (Figs. 6d, 7c, and 7d). During the rainy season, the absolute value of the suction stress of the mollisols in Gully No. II was lower than that of Gully No. I (Fig. 11a). The smaller absolute values of the suction stress for the mollisols of Gully No. II during the snowmelt season (Figure 11b). Moreover, the smaller suction stress in the snowmelt season may have resulted in strong erosion on the slope of Gully No. II, as shown in Fig. 4.

As the hydrological process of the head cut area is closely related to channel bed erosion, the hydromechanical response influences slope stability. It is important to analyze the possible relationship between the erosion per unit area on the channel bed, soil water storage, and erosion of a steep slope with suction stress. In general, a high absolute value of the suction stress is associated with strong cohesive forces between the soil particles, which is a sign of stability. In contrast, a low absolute value of suction stress suggests a higher potential for slope failure. Therefore, the relationship between the absolute value of the suction stress and erosion per unit area could be negative. Fig. 11c shows the reciprocal relationship between the suction stress and erosion per unit area of the slope. The empirical relationship indicates that gravitational mass wasting occurred on the slope, and the permanent gully expanded when the suction stress remained relatively low for a prolonged period, particularly at approximately 5.6 kPa for the study area.



Erosion of the channel bed is closely related to runoff discharge during erosive rain events. During erosive rain
events, the amount of stored soil water decreases runoff amount and intensity. The less rainwater stored during
erosive rain events, the higher the runoff amount, or the more intensive the channeled flow. Therefore, the
relationship between the soil water storage and erosion per unit area of the channel bed could be negative. Fig. 11d
shows the reciprocal relationship between erosion per unit area of the channel bed and soil water storage. It indicates
that excessive rainwater in erosive rain events could create intensified channeled flow to erode the channel bed if
the stored water in the mollisols reached a threshold, such as 139.3 mm in this study area.

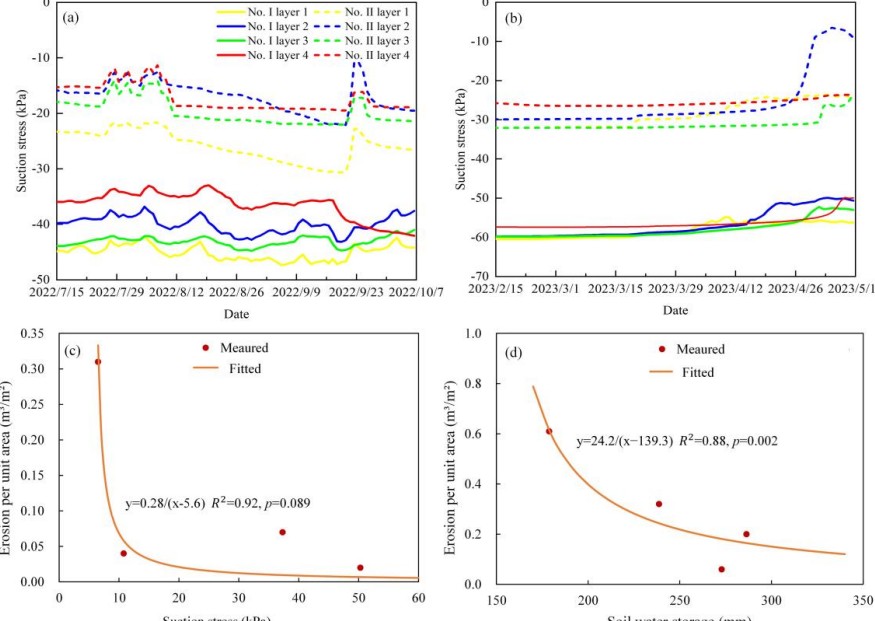


**Fig. 11.** Relationship between hydrology and the hydromechanical state with the erosion per unit area over
approximately 3 months. **(a)** Suction stress during the rainy season. **(b)** Suction stress during the snow melting
season. **(c)** erosion per unit area on the slope decreases with suction stress. **(d)** erosion per unit area on the
channel bed decreases with soil water storage amount. The time for the monitored rainy and melting seasons
were 111 d and 97 d.
**5 Discussion**
The physical processes of permanent gully development can be categorized into gravitational mass wasting on
steep slopes and sediment delivery on channel beds (Montgomery and Dietrich, 1992; van Beek et al., 2008; Luffman
et al., 2015). Traditionally, most studies on gully erosion have focused on soil loss owing to water erosion and piping.
Soil loss estimation is typically determined by several primary factors, such as the upslope contributing area,
topographic conditions, erosive rainfall factors, and land use conditions (Li et al., 2015; Xu et al., 2017; Wang et al.,
2021; Tang et al., 2022). The physical mechanics of bed erosion and slope erosion are different, making it
challenging to accurately predict soil loss on steep slopes. The gravitational mass wasting process on a slope differs
from that of rainfall-induced shallow landslides, especially for those without failure planes (Poesen et al., 1998; Guo
et al., 2020). However, they share similarities, such as a decrease in soil strength due to water infiltration (Guo et al.,
2019). Thus, a thorough mechanical analysis is necessary to understand the physical processes of gravitational mass





wasting on the slope and sediment delivery on the channel bed.
This study thoroughly investigated the effects of hydrological factors and hydromechanical properties on soil
loss on both slopes and channel beds. Mass failure on the hillslopes was governed by suction stress. Meanwhile,
erosion on the channel beds was influenced by the soil water storage or runoff amount. Therefore, hydrological
factors related to soil water storage and drainage were analyzed (Fig. 10), along with volumetric changes at various
rain events and snow melting stages (Fig. 8). In this study, we also investigated the hydromechanical properties and
pore water pressure at a given confining stress (Table 2 and Fig. 5), the relationship between the saturation degree
and suction stress (Fig. 6), and estimated the suction stress variation during the rainy and snow melting seasons (Figs.
11a and 11b). Field observations revealed two permanent gullies with distinct erosion on the slope and gully beds.
Gully No. II shows signs of head cut disruption, in contrast to Gully No. I, resulting in disparities in erosion per unit
area for both seasons and sites. The hydromechanical properties of the mollisols are distinct between the two gullies,
directly affecting water movement. This is evident from the increase in pore water pressure, dissipation ratio, and
proxy. In the head cut of Gully No. II, the mollisols were significantly disturbed, and the soil mass had higher
permeability and lower suction stress at a given saturation degree. This finding indicates more active water
infiltration compared to Gully No. I was triggered by changes in the soil's capacity to store and release water and
the higher volumetric water content increasing ratio. Therefore, the head cut area of Gully No. II underwent more
aggressive hydrological processes. Additionally, the observed rainfall amount of 139.3 mm in this study was smaller
than 177 mm proposed by Tang et al. (2023). This could be explained by the different capacities for plant interception
and depression detention during the rainy season.
The soil water storage and drainage capacity at the head cut considerably influenced soil loss. Although this
study focused primarily on soil water storage and its impact, runoff was not addressed. The soil water storage and
runoff depth were approximately equal to the rainfall depth from the perspective of water balance. Consequently,
the erosion per unit area of the channel bed was inversely proportional to the soil water storage, as shown in Fig.
11d. Some researchers have identified factors leading to the erosion of mass failures on steep slopes, such as long-
duration storms (Xu et al., 2020), initial soil moisture in the pre-winter season (Wen et al., 2024), presence of tensile
crack morphology (Zhou et al., 2023) and heaving and thawing (Thomas et al., 2009). The head cut of Gully No. II
has a high level of disturbance, which may result in higher permeability, quicker water pressure response, and higher
soil moisture during either the rainy or snowmelt seasons. Meanwhile, the soil suction stress was lower, and slope
erosion was more intense than that of Gully No. I. The distance between the two gullies was only 1.4 km and the
climatic conditions were similar. Therefore, soil properties may be the dominant intrinsic factors governing soil loss
on gully slopes.
Long-term saturation during the snowmelt season provides sufficient water infiltration and low suction stress.
Therefore, the highest erosion per unit area occurred in the snowmelt season, but not in the rainy season (Fig. 11c).
This was because of the longer duration of snowmelt compared to rain events (Figs. 7a and 7b). Dong et al. (2011)
revealed that a critical mass water content for gravitational mass wasting ranged from 31.0% to 33.8%,
corresponding to a volumetric water content of 39.0% to 48.0% for the soil mass and a suction stress of 11.0 kPa.
This showed that the direct-shear apparatus limited the ability to differentiate between the effective cohesion and
suction stress contributions to total cohesion. As shown in Fig. 10b and supported by Xu et al. (2020), the high soil
water storage during the snow melting season in Gully No. II (Fig. 9a) and long-term water infiltration can result in
lower suction stress and higher erosion per unit area. This suggests a potentially reciprocal relationship between the
absolute suction stress and erosion per unit area. As shown in Fig. 10, the accuracy of the two empirical equations
can be improved by incorporating data from additional monitoring sites or extending the study period to cover
multiple rainy-snow seasons.





## 6 Conclusions

Permanent gully development is a hydrogeomorphic phenomenon, and its physical mechanics can be attributed to the hydrological and hydromechanical responses of the head cut. In the mollisol region of Northeast China, numerous studies on gully development have focused on soil loss in response to rainfall or snow depth. However, to date, relatively few studies have addressed the physical mechanics of gravitational mass wasting. This study has provided a complete analysis of soil loss on steep slopes and channel beds in two permanent gullies according to hydrological processes, such as infiltration, soil water storage, and drainage, and hydromechanical responses, such as changes in suction stress levels. The following conclusions were drawn:

(1) Mollisols in the head cut areas of Gully No. II exhibited a higher permeability than Gully No. I. This can be attributed to the elevated ratio and proxy for pore water pressure rise and dissipation. The TRIM test results confirmed that the saturated mollisols in the Gully No. II drain faster than Gully No. I, owing to their higher air entry pressure and saturated hydraulic conductivity during the wetting and drying cycles.

(2) The head cut area of Gully No. II exhibited more intense hydrological processes than Gully No. I. This could be explained by the higher ratio of soil moisture increase observed during the four rain event types and three snow melting stages. Soil water storage in Gully No. II experienced greater fluctuations during torrential rains and rainstorms. Overall, the absolute suction in Gully No. II remained lower than that in Gully No. I, potentially triggering greater erosion on the steep slopes.

(3) The relationships between erosion per unit area on the steep slope and channel bed were analyzed for the suction stress and soil water storage. Our findings indicate that low suction stress and high soil water storage can lead to increased gravitational mass wasting while reducing erosion on the channel bed. The two empirical relationships and their efficiency can be improved by incorporating data from ongoing monitoring efforts to enhance the prediction of future soil loss.

### Acknowledgements

All authors declare that no conflict of interest exists. This work was study was supported by the National Key Research and Development Program (Grant No. 2021YFD1500700). The authors extend their gratitude to the colleges at the Jiusan Soil and Water Conservation Experimental Station, Beijing Normal University, for their help during field investigations.

### Code/Data availability

Any readers can contact Prof. Chao Ma as the corresponding author is willing to share the raw/processed data.

### Author contributions

Prof. Ma launched this work based on his skills in gravitational mass-wasting and unsaturated soil mechanics, and proposed the idea-ology of hydrology and hydro-mechanical condition in analyzing the gravitational mass-wasting. Under the guidance of Prof. Ma, Mr. Dongshuo Zheng and Shoupeng Wang finished indoor tests of soil strength and hydraulic-mechanical properties. Prof. Zhang helped determine the field observation sites. Dr. Dong gave insightful comments. Dr. Jie Tang and Yanru Wen provided the research progress about the gravitational mass-wasting on gully expansion in the study area.

### Competing interests

All authors have declared that there were no conflicts of interests and competing interests.



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
