# Peer review of "Understanding soil loss in mollisol permanent gully head cuts through hydrological and hydromechanical responses"

_Hydrology and Earth System Sciences, 2024_

## Author Comment (AC1)

**Comment from RC1 and the replies**

To reviewer:

Thanks for your important comments for this work, your good comments help us improve the quality of this manuscript a lot, such as the clear explanation of the contribution of last figure, high soil loss in channel bed in snow-melting season. We made a point-to-point replies for your comments.

**Comment 1:** The title and objective of this study did not show the key important finding of this study, I suggest the author to make the clear progress in these key parts including the hydromechanical properties.

**Replies:** Done.

Firstly, the title in the previous manuscript does seem to too regular and does not show the important finding of this work. This main contribution of this work, as the comment 2, lies in that the soil loss on bank closely relate to the role of suction stress and the soil loss in channel bed relate to the storied water. Besides, the soil water storage does not equal to the event rainfall amount, but partially from the initial soil water. This finding sufficiently illustrates that the antecedent precipitation or initial soil moisture within the soil plays an important role in soil loss on both bank slope and channel bed. Therefore, we decided to revise the previous title into "**Understanding soil loss in mollisol permanent gully head cuts by hydrological and hydro-mechanical response**". The revised title could reflect the clear progress, including the soil loss, hydromechanical properties, and water storage.

Secondly, the objective of this work (as written in the abstract, final paragraph in the Introduction part and the Discussion part) should strengthen the important finding of this study. Therefore, we revised the three parts following your suggestion. In the revised manuscript, the three parts were revised as:

**Abstract**: During permanent gully development, soil losses on steep slope and in channel bed are typically driven by the hydromechanical response and water storage within the soil mass, while such knowledge have been neglected in previous studies of gully erosion in the mollisol region of Northeast China. In this study, erosion intensities during the 111 d of the rainy season and 97 d of the snow melting season were analyzed with respect to soil water storage and drainage capacity, soil suction stress, supported by the monitoring results of soil moisture, temperature, and precipitation and experimental analysis of soil hydromechanical properties. Under the same confining stress, the mollisols in the interrupted head cut of Gully No. II increased more rapidly and dissipated pore water pressure more than at the uninterrupted head cut of Gully No I. The combination of the soil water characteristic curve and the hydraulic conductivity function indicates that the mollisols of Gully No. II had a lower air entry pressure and higher saturated hydraulic conductivity during the wetting and drying cycles than Gully No. I. The head cut area of Gully No. II exhibited rapid response of water infiltration and drainage, and high soil water storage capacity. The absolute suction stresses within the mollisols of Gully No. II was lower than that in Gully No. I, which could lead to high erosion per unit of steep slope area. Importantly, gravitational mass wasting on

steep slopes is closely related to soil suction stress and we observed a correlation between erosion per unit gully bed area and the soil water storage. Therefore, it is more important to predict the soil loss in permanent gully from both soil water storage and the hydromechanical response of soil mass, other than sole rainfall amount.

**Final paragraph in the Introduction part:** Soil loss from gravitational mass wasting on the steep slope of a permanent gully is poorly understood in the MEC. To date, relatively few studies have addressed its relationship with the hydrological and hydro-mechanical response of the soil mass. This work has focused on how the monitored soil water change and the suction stress affect soil loss during the rainy and melting seasons in the head cuts of two permanent gullies, where one head cut experiences no human activity, whereas the other does. Soil loss in the head cut area during the rainy and melting seasons was observed. The differences in the physical properties of the mollisols, such as pore water pressure dissipation at a given confining stress, the soil water characteristic curve (SWCC), and the hydraulic conductivity function (HCF), were compared. The soil loss per unit area on the steep slope and gully bed was analyzed for the soil water storage, drainage and the soil suction stress, respectively. The objective of this study mainly exhibits the relationship between soil loss intensity on steep slope and hydro-mechanical response of the soil mass, and the intensity in channel bed with water storage.

**The discussion part:** Commonly, the gully bed erosion rates mainly depend on runoff intensity, and some study found that the hydraulics of runoff in the rainy season was significantly higher than the snow melting runoff. However, some studies also proved that gully head may retreat faster in the snow-melting season than in the summer (Wu et al., 2008; Hu et al., 2009). In fact, the accumulated snowfall depth during the monitoring duration in this work was high up to 49.6 mm, which was far more than the average snow depth of 30 mm. Besides, the snowfall was melted all during 3 to 10 May 2023. Therefore, heavy snowfall during the winter 2022 and early spring 2023 and the intensive melting may result in the high soil moisture, intensive runoff to cause strong bed erosion.

**Comment 2:** Figure 11 (last figure) is the key finding and main contribution in the study domain of gully erosion, as it clearly clarifies the role of suction stress of storied water on soil loss from slope and gully bed respectively. In predicting the soil loss, the soil water storage (unit: mm) couldn't equal to the event rainfall amount, but partially from the initial soil water. Therefore, I suggest the authors strength the application of the results of figure 11, or give some clear explanations.

**Replies:** Good suggestion here.

We should give a clear explanation to the finding in fig. 11.

In the final paragraph of Discussion. We added a paragraph to give clear explanation:

**The added part is**: The result in Figs. 11c and 11d is a key finding and main contribution in the study domain of gully erosion, as it clearly clarifies the role of suction stress of storied water on soil loss from slope and gully bed respectively. It also tells the truth that the soil water storage couldn't equal to the event rainfall amount, but

partially from the initial soil water. In other words, the effect of antecedent precipitation would be assessed in predicting soil loss as it closely relates to the soil water and generate indirect influence on the runoff generation and intensity. Importantly, the ideology of this work adopts the theory frame that the soil loss at the steep slope occur by the mechanism of bank slope stability and the loss in the gully bed occur on condition that the balance between the shear force from runoff water to soil erodibility. Therefore, it is more important to predict the soil loss in permanent gully from both soil water storage and the hydromechanical response of soil mass, other than sole rainfall amount.

**Comment 3:** Some figures are not clearly show the results: Fig 2, gully No II, the widths decreased from 2 to 3, which was confusion to me.
**Replies:** It's true.

Fig. 1b and 1c clearly shows the planform of the gully area. The shape of the gully area is odd-looking as the head area of gully No. II suffer from excavator interruption. However, the planform of the gully No. II common-looking.

**Comment 4:** Figure 3, normally the gully bed erosion rates was mainly caused by the runoff scouring, and many previous studies proved that the hydraulics of runoff in the rainy season significant higher than the snow melting runoff. So how do you explain the extremely high erosion rates of gully bed 1 in snowmelt season?
**Replies:** Good comments here.

The thicker snowfall depth over average values and longer snow-melting resulted in the high soil loss.

In this work, we claimed that long-term saturation during the snowmelt season provides sufficient water infiltration and low suction stress. Therefore, the highest erosion per unit area occurred in the snowmelt season, but not in the rainy season (Fig. 11c).

However, the extremely high erosion rates of gully bed No. I was found in snowmelt season. This finding can be explained by the heavy snow events with accumulative snowfall depth up to 49.6 mm in the winter of 2022 and the early spring of 2023, which was far more than the average snow depth of 30 mm. The thicker snow depth over average depth means that more soil loss would be eroded, which can be exemplified by the results of Tang et al. (2021). Tang et al. (2021) developed an empirically-based formula to estimate the soil loss during snow melting season:

$$V=0.102\times SN^{1.104}\times SR\times A^{0.9}\times T^{3.0}\times BD$$

where V is the soil loss, SN is snow depth, SR is a dimensionless snow redistribution factor, A is the upslope contributing area, BD is soil bulk density. At given permeant gully, high soil loss must need strong snowfall. As the snowfall depth was up to 49.6 mm in the winter of 2022 and the early spring of 2023, which was far more than the average snow depth of 30 mm.

[Figure]

**The snowfall distribution during the winter 2022 and the early spring 2023**

Another reason for the high extremely high erosion rates of gully bed No. I in snowmelt season lies in that the longer duration of snow-melting than the storm duration (as shown in fig. 7). The intensive snow melting mainly occurred during 3 and 10 May 2023 (about 7 days). However, the duration of storm was far less than the time-lasting of snow-melt.

[Figure]

**The snow-melting duration in the early spring 2023**

**In the revised manuscript, we added a paragraph (Discussion part) to explain this finding**: Commonly, the gully bed erosion rates mainly depend on runoff intensity, and some study found that the hydraulics of runoff in the rainy season was significantly higher than the snow melting runoff. However, some studies also proved that gully head may retreat faster in the snow-melting season than in the summer (Wu et al., 2008; Hu et al., 2009). In fact, the accumulated snowfall depth during the monitoring duration in this work was high up to 49.6 mm, which was far more than the average snow depth of 30 mm. Besides, the snowfall was melted all during 3 to 10 May 2023. Therefore, heavy snowfall during the winter 2022 and early spring 2023 and the intensive melting may

result in the high soil moisture, intensive runoff to cause strong bed erosion.

**Comment 5:** Fig 8.a, the bar of "rainy events" is not clear, and in my opinion, showing the average rainfall of each rainfall grade can better reflect the impact on VWC.
**Replies:** Done.

Thanks for your patience and recommendations here. It greatly improves the quality of our text. As you say, showing the average rainfall of each rainfall grade truly can better reflect the impact on VWC, So we made the change as you suggested. Thereinto, The average rainfall during light rain events is 2.24 mm, average rainfall during moderate rain events is 17.22 mm, the average rainfall during torrential rain events is 32.91 mm, the average rainfall during rainstorm events is 50.87 mm. We've made changes to the Fig 8 (a).

[Figure]

The previous fig. 8

**The revised fig. 8**

**Comment 6:** Some important references need be cited, e.g. Hu Gang et al., (2007, 2009), Wu Yongqiu et al., (2008).
**Replies:** Sincerely thanks for your good suggestion here.

Thank you for sharing such important documents about the soil loss in the mollisol region of Northeast China.

We also find important evidences from them that the they all claimed that: it is remarkable for freeze-thaw erosion in the black soil area of NE China (Hu et al., 2009), gully heads retreated faster in the spring freeze-thaw period than in the summer (Wu et al., 2008), the erosion by freeze thawing and snowmelt accounts for a large percent (Hu

et al., 2007).

In the revised manuscript, we wrote sentences (in the middle part of the third paragraph, Introduction part) to summarize their important results to support my ideology of this work, and the three documents improved the structure of this manuscript a lot.

**The added sentences are** "Note that some studies proved that the soil loss during snow-melting season remarkably accounts for a large percent (Hu et al., 2007 and 2009), and gully heads retreated faster than in the summer (Wu et al., 2008). Currently, the hydrological processes near the head cut and the hydromechanical response of mollisols to water infiltration in the two seasons have never been documented.

The three works are:

Hu, G., Wu, Y., Liu, B., Yu, Z., You, Z., and Zhang, Y.: Short-term gully retreat rates over rolling hill areas in black soil of Northeast China, Catena, 71, 321-329, https://doi.org/10.1016/j.catena.2007.02.004, 2007.

Hu, G., Wu, Y., Liu, B., Zhang, Y., You, Z., and Yu, Z.: The characteristics of gully erosion over rolling hilly black soil areas of Northeast China, J Geogr Sci., 19, 309-320, https:// 10.1007/s11442-009-0309-4, 2009.

Wu, Y., Zheng, Q., Zhang, Y., Liu, B., Cheng, H., and Wang, Y.: Development of gullies and sediment production in the black soil region of northeastern China, Geomorphology, 101, 683-691, https://doi.org/10.1016/j.geomorph.2008.03.008, 2008.

Tang, J., Liu, G., Xie, Y., Duan, X., Wang, D., and Zhang, S.: Ephemeral gullies caused by snowmelt: A ten-year study in northeastern China, Soil Tillage Res., 212, 105048, https://doi.org/10.1016/j.still.2021.105048, 2021.

---

## Author Comment (AC2)

**Comments and replies to Anonymous Referee #2**

Brief comments: This work exhibits the important of hydro-mechanical properties on the soil loss in channel bed and on steep slope in the mollisol region of Northeast China. In comparison with previous studies on gully erosion at monitoring sites, the work clearly shows the some unknow, but extreme important aspects in the gully development that previous studies haven't been addressed, including the headwater hydrology, suction stress, and their influence on the soil loss. In lots of studies on gully development, most works either solely analyze the channel head retreat rate, gully area expansion, gully area-volume, etc., or develop soil loss equation on rainfall index, they cannot combine the observed soil loss on steep sidewall, with the hydrological factors, not to mention the hydro-mechanical status in the headwaters. Therefore, I recommend the work could be accepted after some minor revisions. Following comments can be considered for the author and may be helpful for the quality improvements for the manuscript.

**Replies**: Thanks for your positive recognition for our study.

This work mainly contributes to know the physical process of permanent gully development in the mollisol region of Northeast China. Traditionally, the studies of permanent gully development focus on the some empirically-based finding, such as the area-volume, channel head retreat rate, and tillage measures on the soil loss. However, there were no studies about the hydrological process and the hydro-mechanical response of head cut. Therefore, this work clearly presents the knowledge gap by analyzing the hydrological and hydro-mechanical response. Besides, our finding also contributes to know about the soil loss problems, e.g., the rain depth of storm event cannot be used in predicting soil loss, as the soil moisture level has a considerable effect on the surface runoff production and the suction stress, which have a close relationship with the erosion on channel bed and the steep bank.

In the final part of Abstract, we also strength our finding using flowing contents: Therefore, it is more important to predict the soil loss in permanent gully from both soil water storage and the hydromechanical response of soil mass, other than sole rainfall amount. In other words, the required water storage capacity to produce runoff intensity and low suction stress would give more accurate results in predicting soil loss in the permanent gully head-cut.

Comment 1: The title or the abstract should highlight the important finding of this work. The methods in "3.4 Soil water storage and drainage" and the figure 11 sufficiently illustrate that the soil loss prediction cannot be from the event rainfall, but from the antecedent precipitation. It is rational and logistical to consider antecedent precipitation in predicting soil loss because the soil water status greatly influent the time, intensity of runoff and the stability of soil on the steep slope. Therefore, I suggest the authors should extend the finding in the discussion part.

**Reply**: Thanks for your insightful comments on the title and abstract of hess-2024-268. Meanwhile, we should strengthen our finding in the discussion part.

Anonymous Referee #1 also give us the same comments on the title and abstract of hess-2024-268. We revised the previous title "Understanding soil loss in two permanent gully head cuts in the mollisol region of Northeast China", into "Understanding soil loss in mollisol permanent gully head cuts by hydrological and hydro-mechanical response". The revised title would be better than the previous one as it highlights the ideology used in this work.

We revised the previous Abstract into "During permanent gully development, soil losses on steep slope and in channel bed are typically driven by the hydromechanical response and

water storage within the soil mass, while such knowledge have been neglected in previous studies of gully erosion in the mollisol region of Northeast China. In this study, erosion intensities during the 111 d of the rainy season and 97 d of the snow melting season were analyzed with respect to soil water storage and drainage capacity, soil suction stress, supported by the monitoring results of soil moisture, temperature, and precipitation and experimental analysis of soil hydromechanical properties. Under the same confining stress, the mollisols in the interrupted head cut of Gully No. II increased more rapidly and dissipated pore water pressure more than at the uninterrupted head cut of Gully No I. The combination of the soil water characteristic curve and the hydraulic conductivity function indicates that the mollisols of Gully No. II had a lower air entry pressure and higher saturated hydraulic conductivity during the wetting and drying cycles than Gully No. I. The head cut area of Gully No. II exhibited rapid response of water infiltration and drainage, and high soil water storage capacity. The absolute suction stresses within the mollisols of Gully No. II was lower than that in Gully No. I, which could lead to high erosion per unit of steep slope area. Importantly, gravitational mass wasting on steep slopes is closely related to soil suction stress and we observed a correlation between erosion per unit gully bed area and the soil water storage. Therefore, it is more important to predict the soil loss in permanent gully from both soil water storage and the hydromechanical response of soil mass, other than sole rainfall amount. In other words, the required water storage capacity to produce runoff intensity and low suction stress would give more accurate results in predicting soil loss in the permanent gully head-cut.".

In the last two paragraphs in the Discussion part, we extended our finding, e.g., the figure 11. The last two paragraphs were revised into "Commonly, the gully bed erosion rates mainly depend on runoff intensity, and some study found that the hydraulics of runoff in the rainy season was significantly higher than the snow melting runoff. However, some studies also proved that gully head may retreat faster in the snow-melting season than in the summer (Wu et al., 2008; Hu et al., 2009). In fact, the accumulated snowfall depth during the monitoring duration in this work was high up to 49.6 mm, which was far more than the average snow depth of 30 mm. Besides, the snowfall was melted all during 3 to 10 May 2023 (Figs. 7a and 7b). Therefore, heavy snowfall during the winter 2022 and early spring 2023 and the intensive melting may result in the high soil moisture, intensive runoff causes strong bed erosion. Long-term saturation during the snowmelt season provides sufficient water infiltration and low suction stress. Therefore, the highest erosion per unit area occurred in the snowmelt season, but not in the rainy season.

Dong et al. (2011) revealed that a critical mass water content for gravitational mass wasting ranged from 31.0% to 33.8%, corresponding to a volumetric water content of 39.0% to 48.0% for the soil mass and a suction stress of 11.0 kPa. This showed that the direct-shear apparatus limited the ability to differentiate between the effective cohesion and suction stress contributions to total cohesion. As shown in Fig. 10b and supported by Xu et al. (2020), the high soil water storage during the snow melting season in Gully No. II (Fig. 9a) and long-term water infiltration can result in lower suction stress and higher erosion per unit area. This suggests a potentially reciprocal relationship between the absolute suction stress and erosion per unit area. The result in Figs. 11c and 11d is a key finding and main contribution in the study domain of gully erosion, as it clearly clarifies the role of suction stress of storied water on soil

loss from slope and gully bed respectively. It also tells the truth that the soil water storage couldn't equal to the event rainfall amount, but partially from the initial soil water. In fact, figure 11 specially illustrate that antecedent soil moisture or precipitation have influence on surface runoff depth and soil loss during the permanent gully expansion in MEC, while this important aspect has been neglected. In other words, the effect of antecedent precipitation would be assessed in predicting soil loss as it closely relates to the soil water and generate indirect influence on the runoff generation and intensity (Sachs and Sarah, 2017; Wei et al., 2017; Schoener and Stone, 2019; Wang et al., 2019). Importantly, the ideology of this work adopts the theory frame that the soil loss at the steep slope occur by the mechanism of bank slope stability and the loss in the gully bed occur on condition that the balance between the shear force from runoff water to soil erodibility. Therefore, it is better way to predict the soil loss in permanent gully from both soil water storage and the hydromechanical response of soil mass, other than sole rainfall amount."

**Comment 2**: I have some issues on the figure 6. Why the lines of wetting and drying process for the same soil are different? Is it because of the air pressure or different tests procedures? Can you give some explanations in the text? This would be helpful for the readers?
**Reply**: Done.

We used the Transient Release and Imbibition Method to obtain the wetting and drying process of the soil. We added sentences in the revised manuscript to give a brief explanation "The reason why the SWCC and HCF of drying process and wetting process are different lies in that water flow from drying process relates to the applied suction level, while the water flow during the wetting process was measured at a positive pressure head (Lu and Godt, 2013)."

**Comment 3**: Some references about the antecedent precipitation on the runoff or soil loss can be considered in citation. In fact, most of the soil loss prediction (such as USL equation) mainly base on the rainfall and runoff factor. The effect of antecedent precipitation has a great influence on the runoff factor.
**Reply**: Good suggestion. You and the Anonymous Referee #1 remined us that important references should be cited in the discussion part. In particular, some references about the antecedent soil moisture could be cited in the text, which can be helpful for manuscript improvement.

We cited four references in the last paragraph in the discussion part. The added sentences are: "In fact, figure 11 specially illustrate that antecedent soil moisture or precipitation have influence on surface runoff depth and soil loss during the permanent gully expansion in MEC, while this important aspect has been neglected. In other words, the effect of antecedent precipitation would be assessed in predicting soil loss as it closely relates to the soil water and generate indirect influence on the runoff generation and intensity (Sachs and Sarah, 2017; Wei et al., 2017; Schoener and Stone, 2019; Wang et al., 2019)."
The four references are:
Wei, L., Zhang, B., and Wang, M.: Effects of antecedent soil moisture on runoff and soil erosion in alley cropping systems, Agr Water Manage., 94, 54-62, https://

doi.org/10.1016/j.agwat.2007.08.007, 2007.

Schoener, G. and Stone, M. C.: Impact of antecedent soil moisture on runoff from a semiarid catchment, J Hydrol., 569, 627-636, https://doi.org/10.1016/j.jhydrol.2018.12.025, 2019.

Wang, F., Tian, P., Guo, W., Chen, L., Gong, Y., and Ping, Y.: Effects of rainfall patterns, vegetation cover types and antecedent soil moisture on run‑off and soil loss of typical Luvisol in southern China, Earth Surf Process Landf., 49, 2998-3012, https://doi.org/10.1002/esp.5871, 2024.

Sachs, E. and Sarah, P.: Combined effect of rain temperature and antecedent soil moisture on runoff and erosion on Loess, Catena, 158, 213-218, https://doi.org/10.1016/j.catena.2017.07.007, 2017.

---

## Author Response (AR1)

Dear Thom Bogaard and the two anonymous reviewers

Thanks for your comments, help and insightful suggestion for improving the quality of the manuscript HESS-2024-268, titled "Understanding soil loss in two permanent gully head cuts in the mollisol region of Northeast China".

After the first reviews process, we continuously revise the manuscript according to requirements of Hydrology and earth system science and previous comments from the editor and the reviewers. In previous submission, both the editor and reviewers pointed us that some important information are missing and we need a thorough revision. Then, we revised and submitted manuscript again.

We already have made a point-to-point revision according to the two anonymous reviewers' comments and replied to them, which can be seen in the replies to RC 1 to 5. Except some great revisions for the title, abstract and the discussion (see following point-to-point revisions), some figures are also improved.

Following aspects show the important revisions:

Comment 1: The title or the abstract should highlight the important finding of this work. The methods in "3.4 Soil water storage and drainage" and the figure 11 sufficiently illustrate that the soil loss prediction cannot be from the event rainfall, but from the antecedent precipitation. It is rational and logistical to consider antecedent precipitation in predicting soil loss because the soil water status greatly influent the time, intensity of runoff and the stability of soil on the steep slope. Therefore, I suggest the authors should extend the finding in the discussion part.

**Revisions**:

We revised the previous title "Understanding soil loss in two permanent gully head cuts in the mollisol region of Northeast China", into "**Understanding soil loss in mollisol permanent gully head cuts by hydrological and hydromechanical response**". The revised title would be better than the previous one as it highlights the ideology used in this work.

We revised the previous Abstract into "**During permanent gully development, soil losses on steep slopes and in channel beds are typically driven by the hydromechanical response and water storage within the soil mass; however, this knowledge has been largely neglected in previous studies of gully erosion in the mollisol region of Northeast China. In this study, erosion intensities during the 111 d of the rainy season and 97 d of the snow-melting season were analyzed with respect to soil water storage, drainage capacity, and soil**

suction stress, supported by monitoring results of soil moisture, temperature, and precipitation, as well as experimental analysis of soil hydromechanical properties. Under the same confining stress, the mollisols in the interrupted head cut of Gully No. II increased more rapidly and dissipated pore water pressure more effectively than those at the uninterrupted head cut of Gully No. I. The combination of the soil water characteristic curve and the hydraulic conductivity function indicated that the mollisols of Gully No. II had a lower air-entry pressure and higher saturated hydraulic conductivity during the wetting and drying cycles than Gully No. I. The head cut area of Gully No. II exhibited rapid water infiltration and drainage response and high soil water storage capacity. The absolute suction stresses within the mollisols of Gully No. II was lower than that in Gully No. I, which could lead to high erosion per unit of steep slope area. Importantly, gravitational mass wasting on steep slopes was closely related to soil suction stress, and we observed a correlation between erosion per unit in the gully bed area and soil water storage. Therefore, it is more important to predict the soil loss in the permanent gully from soil water storage and the hydromechanical response of soil mass, other than sole rainfall amount. In other words, the required water storage capacity to yield runoff intensity and low suction stress would predict soil loss in the permanent gully head cut more accurately.".

In the last two paragraphs in the Discussion part, we extended our finding, e.g., the figure 11. The last two paragraphs were revised into "Commonly, the gully bed erosion rates mainly depend on runoff intensity, and some studies reported that the runoff hydraulics in the rainy season were significantly higher than the snow-melting runoff. However, additional studies proved that gully heads may retreat faster in the snow-melting season than in the summer (Wu et al., 2008; Hu et al., 2009). The accumulated snowfall depth during the monitoring duration in this study was high, up to 49.6 mm, which was far more than the average snow depth of 30 mm. Besides, the snowfall melted from 3 to 10 May 2023 (Figs. 7a and 7b). Therefore, heavy snowfall during the winter of 2022 and early spring of 2023 and the intensive melting may result in high soil moisture and intensive runoff, ultimately causing substantial bed erosion. Long-term saturation during the snowmelt season provides sufficient water infiltration and low suction stress. Therefore, the highest erosion per unit area occurred in the snowmelt season but not in the rainy season."

Dong et al. (2011) revealed that a critical mass water content for gravitational mass wasting ranged from 31.0% to 33.8%, corresponding to a volumetric water content of 39.0% to 48.0% for the soil mass and a suction stress of 11.0 kPa. This showed that the direct-shear apparatus limited the ability to differentiate between the effective cohesion and suction stress contributions to total cohesion. As shown in Fig. 10b and

supported by Xu et al. (2020), the high soil water storage during the snow-melting season in Gully No. II (Fig. 9a) and long-term water infiltration can lower suction stress and higher erosion per unit area. This suggests a potentially reciprocal relationship between the absolute suction stress and erosion per unit area. The result shown in Figs. 11c and 11d are key findings and main contributions in the study domain of gully erosion, as they clarify the role of suction stress of storied water on soil loss from steep slopes and gully beds, respectively. Our results also imply that the soil water storage may not equal the amount of rainfall from the event, but instead partially derives from the initial soil water. Figure 11 illustrates that antecedent soil moisture or precipitation substantially influences surface runoff depth and soil loss during the permanent gully expansion in MEC, while this critical aspect has been neglected in previous study. In other words, the effect of antecedent precipitation should be assessed in predicting soil loss as it closely relates to the soil water and indirectly influences the runoff generation and intensity (Sachs and Sarah, 2017; Wei et al., 2017; Schoener and Stone, 2019; Wang et al., 2019). Notably, the theoretical framework underlying this work is that the soil loss at steep slopes occurs through the mechanism of bank slope stability, and the loss in gully beds occurs due to the balance between the shear force from runoff water and soil erodibility. Therefore, it is preferrable to predict soil loss in the permanent gullies from soil water storage and the hydromechanical response of soil mass, rather than solely from rainfall amount."

Comment 2: Some references about the antecedent precipitation on the runoff or soil loss can be considered in citation. In fact, most of the soil loss prediction (such as USL equation) mainly base on the rainfall and runoff factor. The effect of antecedent precipitation has a great influence on the runoff factor.

Revisions: We cited four references in the last paragraph in the discussion part. The added sentences are: "Figure 11 illustrates that antecedent soil moisture or precipitation substantially influences surface runoff depth and soil loss during the permanent gully expansion in MEC, while this critical aspect has been neglected in previous study. In other words, the effect of antecedent precipitation should be assessed in predicting soil loss as it closely relates to the soil water and indirectly influences the runoff generation and intensity (Sachs and Sarah, 2017; Wei et al., 2017; Schoener and Stone, 2019; Wang et al., 2019)."

The four references are:

Wei, L., Zhang, B., and Wang, M.: Effects of antecedent soil moisture on runoff and soil erosion in alley cropping systems, Agr Water Manage., 94, 54-62, https://

doi.org/10.1016/j.agwat.2007.08.007, 2007.

Schoener, G. and Stone, M. C.: Impact of antecedent soil moisture on runoff from a semiarid catchment, J Hydrol., 569, 627-636, https://doi.org/10.1016/j.jhydrol.2018.12.025, 2019.

Wang, F., Tian, P., Guo, W., Chen, L., Gong, Y., and Ping, Y.: Effects of rainfall patterns, vegetation cover types and antecedent soil moisture on run‑off and soil loss of typical Luvisol in southern China, Earth Surf Process Landf., 49, 2998-3012, https://doi.org/10.1002/esp.5871, 2024.

Sachs, E. and Sarah, P.: Combined effect of rain temperature and antecedent soil moisture on runoff and erosion on Loess, Catena, 158, 213-218, https://doi.org/10.1016/j.catena.2017.07.007, 2017.

Some figure revisions:

[Figure]

**Fig. 4.** Differences in the erosion per unit area for the gully bed and slope

[Figure]

**Fig. 8.** Volumetric water content increasing ratio in snow-melting ratio and the rainy season. (**a**) Rate of increase in VWC at varied rain events. (**b**) Rate of increase in VWC at three stages of temperature increase.